# Occupational Exposure Assessment of the Static Magnetic Field Generated by Nuclear Magnetic Resonance Spectroscopy: A Case Study

**DOI:** 10.3390/ijerph19137674

**Published:** 2022-06-23

**Authors:** Valentina Hartwig, Carlo Sansotta, Maria Sole Morelli, Barbara Testagrossa, Giuseppe Acri

**Affiliations:** 1Institute of Clinical Physiology IFC-CNR, 56100 Pisa, Italy; valeh@ifc.cnr.it; 2Department of Biomedical, Dental, Morphological and Functional Imaging Sciences, University of Messina, 98122 Messina, Italy; sansotta@unime.it (C.S.); btestagrossa@unime.it (B.T.); 3Fondazione Toscana Gabriele Monasterio, 56100 Pisa, Italy; msmorelli@monasterio.it

**Keywords:** occupational exposure, RISK evaluation, NMR, AIMDs, MR safety

## Abstract

Magnetic resonance (MR) systems are used in academic research laboratories and industrial research fields, besides representing one of the most important imaging modalities in clinical radiology. This technology does not use ionizing radiation, but it cannot be considered without risks. These risks are associated with the working principle of the technique, which mainly involves static magnetic fields that continuously increase—namely, the radiofrequency (RF) field and spatial magnetic field gradient. To prevent electromagnetic hazards, the EU and ICNIRP have defined workers’ exposure limits. Several studies that assess health risks for workers and patients of diagnostic MR are reported in the literature, but data on workers’ risk evaluation using nuclear MR (NMR) spectroscopy are very poor. Therefore, the aim of this research is the risk assessment of an NMR environment, paying particular attention to workers with active implantable medical devices (AIMDs). Our perspective study consisted of the measurement of the static magnetic field around a 300 MHz (7 T) NMR research spectrometer and the computation of the electric field induced by the movements of an operator. None of the calculated exposure parameters exceeded the threshold limits imposed by legislation for protection against short-term effects of acute occupational exposure, but our results revealed that the level of exposure exceeded the action level threshold limit for workers with AIMD during the execution of tasks requiring the closest proximity to the spectrometer. Moreover, the strong dependence of the induced electric field results from the walking speed models is shown. This case study represents a snapshot of the NMR risk assessment with the specific goal to increase the interest in the safety of NMR environments.

## 1. Introduction

Magnetic resonance (MR), before finding wide application in medicine as a diagnostic technique, was widely used in (laboratory) basic research and for the assessment of food quality, and in the study of organic molecules through the interpretation of nuclear magnetic resonance (NMR) spectra. In particular, NMR spectroscopy provides detailed information about the structure, dynamics, reaction state, and chemical environment of molecules [1,2,3].

In general, diagnostic magnetic resonance imaging (MRI) and NMR spectroscopy use the same working principle, which involves three steps: the polarization of the magnetic nuclear spins when the sample (or human body) is placed in a static magnetic field (B_0_); the irradiation of the sample by a radiofrequency (RF) magnetic field (B_1_), whose frequency is close to the Larmor frequency; and the application of spatial magnetic field gradients to the localization of the spins [4,5,6,7].

The possibility that some hazards may be associated with MR devices has been of concern, considering the rapid development of technologies, in particular, MRI [8,9,10,11].

The main risks for workers are associated with static and spatially heterogeneous magnetic fields and, occasionally, RF fields. Moving around the magnet, the worker is also exposed to a time-varying magnetic field that induces electrical currents in the body. When the currents are strong enough, they can cause effects on the central nervous system (CNS) and stimulate peripheral nerves. Recent studies have reported transient symptoms induced by workers’ movements [12,13,14,15].

Occupational exposure to the RF field is low since the field falls off rapidly outside the transmit coil. However, workers could be exposed to the RF field during MRI interventional procedures to levels similar to those experienced by patients or volunteers undergoing the procedure, but this kind of exposure is not involved in the case of those working with NMR spectrometers, because of the active shield of the spectrometer.

Due to this concern, the EU and ICNIRP have defined workers’ exposure limits to the risks arising from physical agents (electromagnetic fields) [16,17,18].

The objective of the ICNIRP guidelines [18] is to prevent peripheral nerve stimulation and to minimize the possibility of transient sensory effects as a consequence of electric fields induced in the human body by movements in static magnetic fields within occupational settings.

In 2014, the ICNIRP published the “Guidelines for Limiting Exposure to Electric Fields Induced by Movement of the Human Body in a Static Magnetic Field and by Time-Varying Magnetic Fields below 1 Hz”, establishing, for controlled exposure conditions, a basic restriction of 1.1 V/m for the peak induced electric field and a reference level of 2.7 T/s for the time derivative of the magnetic flux density (*dB/dt*) [18].

Various studies assessing the health risk for MRI workers have been published [19,20,21,22,23]. Workers’ exposure to static magnetic fields and their movements in fringe fields have been evaluated and discussed in sites that used MRI scanners ranging from 0.25 T up to 3.0 T [24,25,26].

To our knowledge, only very few studies have been conducted on workers’ risk evaluation in relation to NMR spectrometers. NMR spectrometers are characterized by a very high static magnetic field (up to 28 T, 1.2 GHz commercially available).

NMR spectrometers are more widespread in universities and research centers, and they are also used by PhD and fellow students, which are considered “scientific users” and not workers. Usually, they are very competent in the scientific application of the technique but are not very sensitive to safety aspects, and they may mistakenly think that the use of NMR spectrometers involves little risk.

In this context, it is essential to educate workers to behave correctly and to control their movements to avoid adverse events.

The aim of this study was to evaluate the static magnetic field around an NMR spectrometer (300 MHz, 7 T) and, starting from these measurements, to provide a theoretical evaluation, made by using a developed software, of the time-varying magnetic fields due to the movement of workers.

In this way, the evaluation of worker exposure, as risk assessment, is carried out by considering the induced electric field |*E*| and the time derivative of magnetic flux density |*dB/dt*| for providing a safe working procedure. Particular attention is paid to workers with active implantable medical devices (AIMDs) as a particularly sensitive risk group that must be protected against the dangers caused by the interference of electromagnetic fields.

## 2. Materials and Methods

The static magnetic field of an NMR spectrometer (Bruker, 300 MHz, 7 T) environment was assessed: Table 1 reports the main characteristics of the spectrometer and the room in which it is installed.

The mapping of the static magnetic field was performed by means of a three-axis Hall magnetometer THM 1176 (Metrolab Instruments SA, Plan-les-Ouates, Switzerland). The consistency of this kind of instrumentation was already evaluated by authors in previous studies [14,27,28,29]. The probe of the magnetometer was placed at five different heights from the ground plane (z = 0 cm, 40 cm, 80 cm, 120 cm, and 160 cm). According to anthropometric tables [30], which contain data on human body size and shape and are the basis upon which all digital human models are constructed, the chosen heights are relative to the following body parts for a 170 cm tall man in a standing position:z = 160 cm head;z = 120 cm thorax–heart;z = 80 cm hip;z = 60 cm hands;z = 40 cm knee.

Moreover, if we consider a worker in a squatting position, for example, to access the lower part of the NMR spectrometer during manual parameter shimming or loading samples, z = 80 cm can be considered as the position of the head.

Each measurement of the static magnetic field was made on a theoretical path traveled by the worker, moving toward the spectrometer from the console and vice versa, with a step of 10 cm. The covered area was between the NMR spectrometer and the operator console placed in the same room.

We measured the modulus of the magnetic field |*B*| at each point: The measurements were repeated three times to check the repeatability and the reproducibility of the measurements, either in unchanged conditions or changing the people assigned to the measure [31]. The gaussmeter was programmed to take instant *B* values every 10 s during a 1 min recording. The magnetic field |*B*| for each of the different measurements was estimated by the means and standard deviations (Mean ± σ_n−1_) of the stored values. We obtained an array of 28 × 5 values for the magnetic field. Starting from these measurements, the static magnetic field 2D distribution on the vertical plane (xz plane) was calculated by fitting the data using an exponential interpolation, on a grid with a resolution of 1 × 1 cm. All calculations were performed with a homemade MATLAB^®^, R2020b (MathWorks, Inc., Natick, MA, USA) script. Then, we calculated the modulus of spatial gradient |dB/ds| with respect to the x- and z-axes.

The ICNIRP guidelines for limiting exposure to electric fields induced by movement of the human body in a static magnetic field [18] indicate basic restrictions in controlled conditions in terms of motion-induced internal electric field strength.

From the knowledge of the 2D distribution of |*B*| on the xz plane, the induced electric field |*E*| was calculated for a trajectory covered by a worker during a routine operation of sample processing.

|*E*| was estimated using the analytical model proposed in the ICNIRP guidelines [18]. Starting from Faraday’s law, which indicates that the induced electric field is directly related to the change in the magnetic flux through the body, and resolving it for a body cross-section perpendicular to the magnetic field, we obtained the following equation [32,33]:(1)$$\left|E\right|=C\left|\frac{dB}{dt}\right|=C\cdot \left(\left|\frac{dB}{ds}\right|\cdot v\right),$$
where |*dB*/*dt*| is the time derivative of the magnetic flux density, |*dB*/*ds*| represents the spatial gradient of the static magnetic field, and *v* is the walking speed of the worker. *C* is a geometric multiplier depending on the size and the shape of the body, as well as the direction and distribution of the magnetic field. The geometric multiplier can be determined considering an elliptical or circular cross-section perpendicular to the magnetic field [34]. In this study, we considered an elliptical section, with a = 0.4 m as the semi-major axial length and b = 0.2 as the semi-minor (*C* = 0.16 Vm^−1^ per Ts^−1^).

The ICNIRP guidelines [18] also set reference levels in terms of |*dB/dt*|: this parameter was calculated from |*E*| using Equation (1).

The walking speed was chosen considering some studies in the literature [25,28,35] that are based on both theoretical consideration and observation of movements of workers in real scenarios. We set some models of walking speed and some values of maximum walking speed, for a linear path along the *x*-axis from the console to the NMR spectrometer and vice versa. Table 2 shows the chosen models and the maximum values of walking speed.

For example, MOD3 is considered a walking speed that increases linearly in the first part of the path, then remains constant and equal to the maximum value (*v*_max_) in the central part of the path, and finally decreases linearly up to the stopping point.

The obtained results of |*E*| and |*dB/dt*| were compared with the ICNIRP exposure limits [18].

## 3. Results

All values relative to magnetic flux density measurements are reported here in mT (1 mT = 10 Gauss).

Figure 1a shows the color map of the spatial gradient |*dB/ds*| in the area of interest between the NMR spectrometer and the console, calculated with respect to the x- and z-axes, from the |*B*| measurements at five different heights from the ground plane (z = 0 cm, 40 cm, 80 cm, 120 cm, and 160 cm). Figure 1b depicts the mean |*B*| values recorded at different heights from the ground plane and standard deviation (SD); in particular, SD is shown as a color gradient.

The result for the xz plane (perpendicular to the ground plane, @y = NMR isocenter) is shown. The red stars in the figure represent points of measurement. By using this map, it was possible to identify the area in which the workers were exposed to a higher spatial gradient of the magnetic field. The maximum value of the spatial gradient in this area was 71.27 mT/m.

Figure 2 shows the modulus of the magnetic flux density |*B*| along a linear trajectory parallel to the *x*-axis starting close to the NMR spectrometer (x = 0 cm) and moving toward the console position (x = 270 cm), at the chosen five different heights with respect to the floor. The action level (AL) specified by the regulations [16,36,37,38] to limit interference with the function of active implantable medical devices (AIMDs) at 0.5 mT is also shown, together with the AL of 3 mT set to limit the projectile risk in the fringe field from strong sources (>100 mT). It is possible to observe that, at z = 120 cm (@thorax–heart), the level of exposure exceeded the AL of AIMDs at a distance of 90 cm from the NMR spectrometer (180 cm away from the console), while at z = 160 cm (@head), the level of exposure exceeded the AL for AIMDs at a distance of 96 cm from the NMR spectrometer (176 cm away from the console). Regarding the AL for the projectile risk, the exposure level exceeded the specified AL at a distance of 225 cm from the console at z = 80 cm (@hip).

Figure 3 shows the calculated induced electric field |*E*| (in mV/m) in the body of a worker walking along a linear trajectory parallel to the *x*-axis, at the chosen five different heights with respect to the floor. Figure 3a is relative to the walking speed model MOD1. Figure 3b concerns the walking speed model MOD2, which includes two sections—the first section with linear acceleration and the second section with a constant speed equal to *v*_max_. Figure 3c illustrates the diagram of the walking speed model MOD3, which includes three sections—first a linear acceleration, followed by a constant speed equaling *v*_max_, and finally, a linear deceleration. Figure 3a–c concern the path starting close to the NMR spectrometer (x = 0 cm) and moving toward the console position (x = 270 cm). Figure 3d shows results relative to the walking speed model MOD2 for a walking path starting from the console position (x = 270 cm) and moving toward the NMR spectrometer (x = 0 cm). In order to show the worst-case scenarios, Figure 3 reports the results for *v*_max_ equaling *v*_3_ = 2 m/s.

Table 3, Table 4 and Table 5 show the peak value of the calculated parameters |*B*|, |*E*|, and |*dB/dt*| for each walking speed models (MOD1, MOD2, and MOD3) and each maximum speed values (*v*_1_, *v*_2_, and *v*_3_) at z = 80 cm, z = 120 cm, and z = 160 cm, respectively.

Peak values of the calculated exposure parameters for the walking speed model MOD2 along the path starting from the console position to the NMR spectrometer (data not reported) were equal to the peak values obtained for the walking speed model MOD1 along the reverse path. This is because, in the proximity of the NMR spectrometer, where the magnetic field is higher, the walking speed is equal to *v*_max_ in both cases.

## 4. Discussion

At present, the literature on occupational exposure in NMR environments is very poor [39,40] mainly when compared with the literature on the exposure characterization for healthcare staff working with magnetic resonance imaging (MRI) scanners [41,42].

Well-established physical mechanisms of interaction between the MRI-related electromagnetic fields and living tissues are responsible for acute and transient effects occurring above threshold exposure levels, which provide the basis for the definition of the exposure limits. However, regarding chronic occupational exposure, available epidemiological and experimental evidence on the potential adverse effects of static magnetic fields has been considered limited and insufficient to reach definitive conclusions [43].

Regarding exposure assessment, there are no standardized procedures to assess occupational exposure in both MRI and NMR environments. The characterization of exposure levels and conditions is strongly necessary mainly in the case of highly specialized workers in NMR-related industrial or research environments, which have very high values of static magnetic field (up to 28 T). This is also because, at the base of the drafting of effective best practices and guidelines, there is a clear understanding of the physical quantities involved.

In order to explore and characterize the levels of occupational exposure to magnetic fields generated by an NMR spectrometer (@ 7T), in this study, we presented a computational tool to estimate the electric field induced by the movements of an operator.

As expected, the data revealed that higher B levels, corresponding to the highest exposure values, were recorded in the vicinity of the spectrometer. This result is similar to the findings reported in [39].

The measurement of |*B*| and the calculation of its spatial distribution and gradient point to high-risk areas, corresponding to the areas with higher magnetic field space gradient, in which the operator should be more careful and move slowly.

None of the calculated exposure parameters exceeded the relative limits reported by legislation in force [18] whereby the considered exposure conditions were compliant with exposure restrictions.

A complete discussion about both controlled and uncontrolled exposure should include implications of higher spectral components: In this case, a weighted peak approach should be used [18]. However, as the PNS-based basic restrictions for all tissues in controlled exposure conditions are constant over a large frequency range, there is no need for spectral weighting [18]. On the other hand, the NMR environment may pose risks or problems to workers with certain implants and other medical devices primarily due to factors that include electromagnetic field interactions. One of the unique concerns of introducing an AIMD into the MR environment is the potential for device malfunction. Exposure to electromagnetic fields may interfere with the electronic components and cause device failures such as performance degradation, loss of function, or unintentional responses. Observing Figure 2, one can infer that the level of exposure exceeded the AL threshold limit for AIMDs (0.5 mT) at about 1 m from the NMR spectrometer. Our results reveal that the workers can be exposed to fields > 0.5 mT, especially during the execution of tasks requiring the closest proximity to the equipment and for heights from the floor at which typical active medical implants could be present.

Another relevant finding is the strong dependence of the induced electric field results from the walking speed models: As can be derived from Table 3, Table 4 and Table 5, all of the exposure parameters were very higher for the walking speed model MOD1, which considers a constant speed along the entire path. This fact implies that there is a need for a deep knowledge of the worker movements to obtain an accurate estimate of the exposure parameters. A constant walking speed model can be used only for the worst-case scenario, but it does not allow for an exhaustive and realistic estimation of the exposure. Moreover, especially for the head and torso, the rotation movements, as well as angular speeds, should also be considered for a complete assessment of the exposure parameters.

Observing Figure 3a–d, it can be inferred that the maximum value of the induced electric field was obtained in all positions close to the NMR spectrometer, especially in the case in which the worker moves away or approaches the NMR at a high speed (Figure 3a,d). As observed in Figure 3b,c, |*E*| had similar trends, despite referring to different walking speed models. This fact can be explained considering that both MOD2 and MOD3 showed an initial acceleration part, followed by a constant speed part. However, in the first part of the worker path, MOD3 included a higher speed than MOD2, so the values of |*E*| for this case were higher up to x = 180 cm. Then, MOD3 ended with a deceleration in the area with a low magnetic field (from x = 180 cm to x = 270 cm), where, by contrast, MOD2 had a high-speed equal to *v*_max_; hence, in this final part, the values of |*E*| for the MOD2 were higher.

In summary, following the indications given in this study relating to the exposure assessment in an NMR environment, it is possible to set some guidelines and best practices for workers, in order to avoid overexposure conditions, especially for workers at particular risk.

## 5. Conclusions

In this prospective study, we evaluated the safety of an NMR spectrometer for workers during their routine activities. In all cases, body and limb exposure levels were below the limits established by the European Directive for protection against short-term effects of acute occupational exposure. However, the results revealed that during workers’ movement, already at a distance of 1 m from the spectrometer, employees were exposed to a static magnetic field > 0.5 mT, the threshold limit for workers with AIMD. The AIMD devices are usually termed as “legacy” systems, considered by FDA and device manufacturers to be a contraindication to MR.

It is necessary to consider that the technological evolution of NMR tends increasingly toward ever-greater static magnetic fields, with increasingly higher working RF, thus entailing a significant increase in the level of risk for workers.

The NMR risk assessment should involve all interested parties—from the employer who certifies, by means of a medical examination, the health suitability to carry out the tasks involving exposure to the magnetic fields of the spectrometer, to workers that should adopt specific behaviors especially when they move near the spectrometer.

This case study represents a snapshot of the NMR risk assessment, and it will need to be implemented and optimized in the future.

Epidemiological evidence on the potential effects of occupational exposure to static magnetic fields is not considered probative of noxiousness. Studies in this research field are scarce. In future research, it is our intention to implement this study by using wearable pocket dosimeters, which means they do not hinder the operators’ movements. In this manner, it will be possible to gather a wide statistical pool of data to assist studies on health hazards and the need to develop behavioral rules for workers, especially when they move near NMR spectrometers, and for workers with AIMD.

The goal of this study was to increase the interest in safety in those environments where it is not implemented, so as to guarantee the safety of workers and all those involved.

## Figures and Tables

**Figure 1 ijerph-19-07674-f001:**
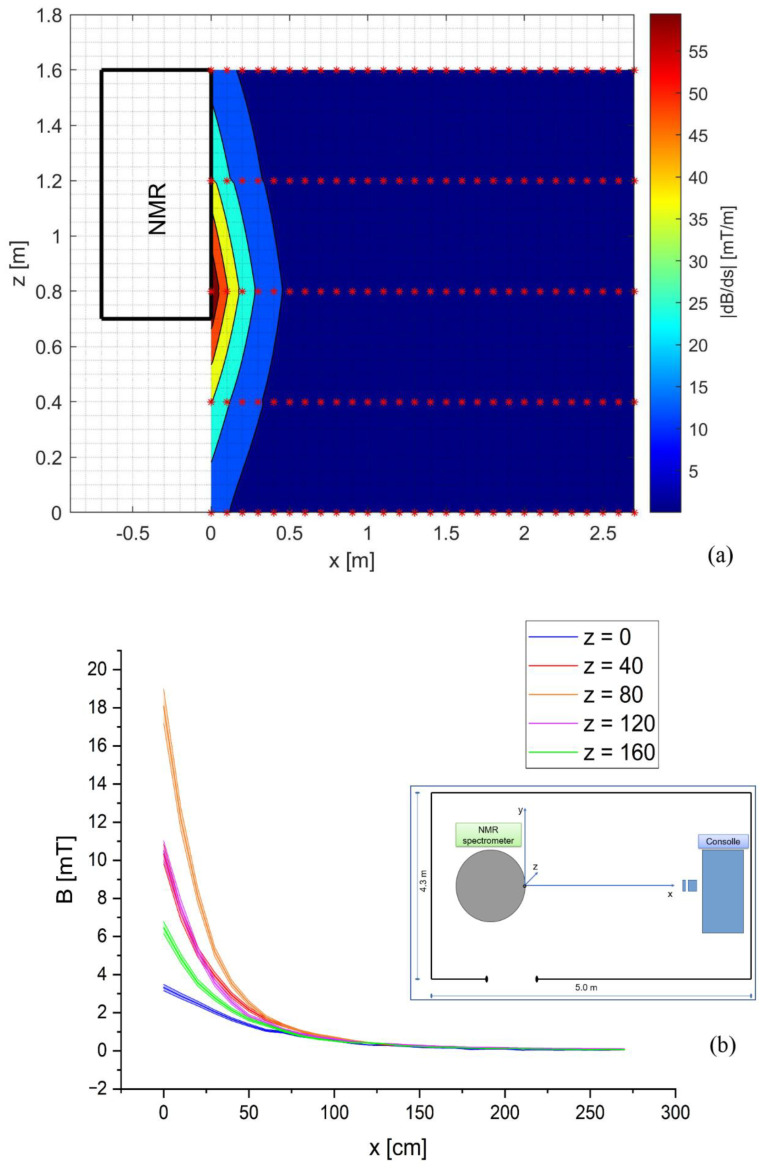
(**a**) Spatial gradient |*dB/ds*| (red stars represent points of measurement of |*B*|); (**b**) the trend of |*B*| values, measured at different heights from ground plane, and the SD depicts as gradient color. The box inside (**b**) explains the room structure, center of coordinate systems, and axis.

**Figure 2 ijerph-19-07674-f002:**
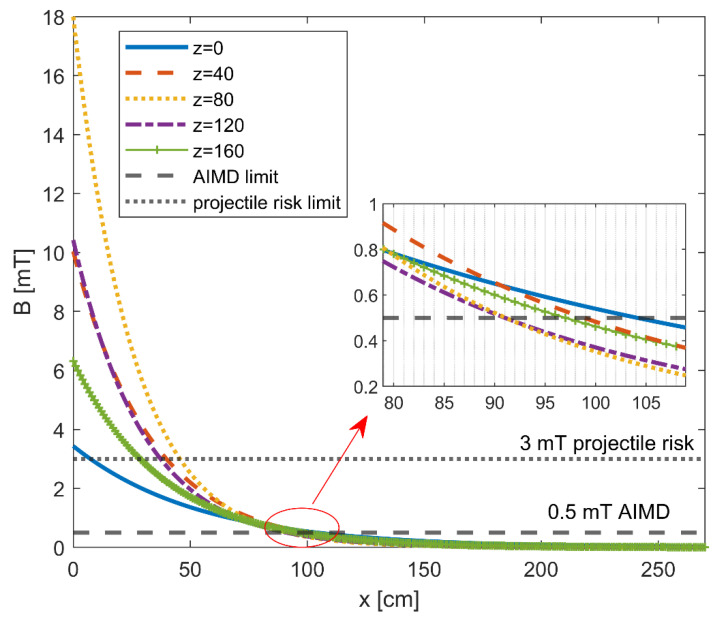
Modulus of the magnetic flux density |*B*| along a linear trajectory parallel to the *x*-axis starting close to the NMR spectrometer (x = 0 cm) and moving toward the console position (x = 270 cm) at different heights with respect to the floor.

**Figure 3 ijerph-19-07674-f003:**
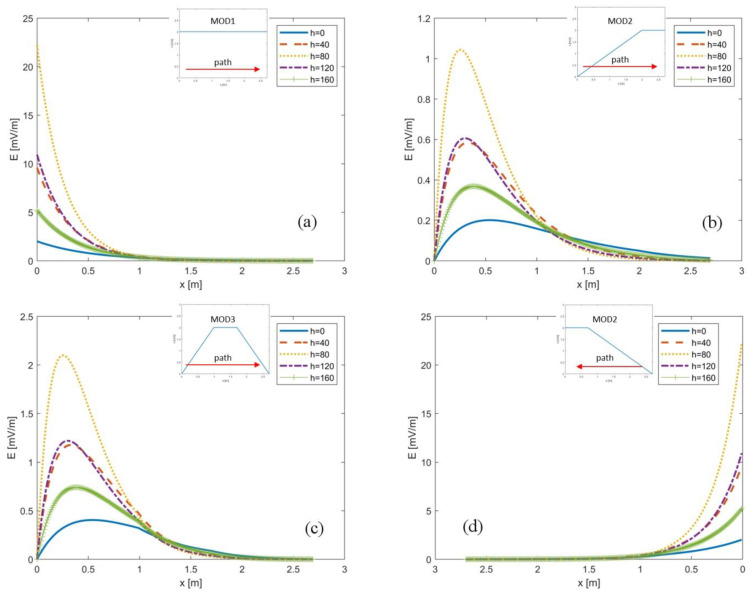
The calculated induced electric field |*E*| (in mV/m) in the body of a worker walking along a linear trajectory parallel to the *x*-axis, where x = 0 m corresponds to a point near the spectrometer and x = 2.7 m is the location of the console: (**a**) walking speed model MOD1, from NMR to console; (**b**) walking speed model MOD2, from NMR to console; (**c**) walking speed model MOD3, from NMR to console; (**d**) walking speed model MOD2, from console to NMR. All the results are for *v*_max_ = 2 m/s.

**Table 1 ijerph-19-07674-t001:** Spectrometer characteristics.

NMR Spectrometer	B_0_ (T)	Frequency (MHz)	Shielding	Room Size (m)
Bruker	7	300	Active	4.30 × 5.00

**Table 2 ijerph-19-07674-t002:** Walking speed models and values.

**MOD 1**	Constant speed = *v*_max_		
**MOD 2**	Linear increase − constant = *v*_max_		
**MOD 3**	Linear increase − constant = *v*_max_ − linear decrease		
** *v* _max_ **	*v*_1_ = 1 m/s	*v*_2_ = 1.6 m/s	*v*_3_ = 2 m/s

**Table 3 ijerph-19-07674-t003:** Peak values of the calculated exposure parameters for z = 80 cm (hip in standing position or head in squatting position).

Peak Value	MOD 1			MOD 2			MOD 3		
	*v* _1_	*v* _2_	*v* _3_	*v* _1_	*v* _2_	*v* _3_	*v* _1_	*v* _2_	*v* _3_
|*B*| mT	18.08	18.08	18.08	18.08	18.08	18.08	18.08	18.08	18.08
|*E*| mV/m	11.11	17.77	22.21	0.52	0.84	1.04	1.05	1.68	2.10
|*dB/dt*| mT/s	69.42	111.07	138.84	3.26	5.22	6.53	6.56	10.50	13.12

**Table 4 ijerph-19-07674-t004:** Peak values of the calculated exposure parameters for z = 120 cm (thorax–heart in standing position).

Peak Value	MOD 1			MOD 2			MOD 3		
	*v* _1_	*v* _2_	*v* _3_	*v* _1_	*v* _2_	*v* _3_	*v* _1_	*v* _2_	*v* _3_
|*B*| mT	10.50	10.50	10.50	10.50	10.50	10.50	10.50	10.50	10.50
|*E*| mV/m	5.47	8.75	10.94	0.31	0.49	0.61	0.61	0.97	1.22
|*dB/dt*| mT/s	34.19	54.70	68.38	1.89	3.03	3.79	3.81	6.09	7.62

**Table 5 ijerph-19-07674-t005:** Peak values of the calculated exposure parameters for z = 160 cm (head in standing position).

Peak Value	MOD 1			MOD 2			MOD 3		
	*v* _1_	*v* _2_	*v* _3_	*v* _1_	*v* _2_	*v* _3_	*v* _1_	*v* _2_	*v* _3_
|*B*| mT	6.46	6.46	6.46	6.46	6.46	6.46	6.46	6.46	6.46
|*E*| mV/m	2.60	4.17	5.21	0.18	0.30	0.37	0.37	0.59	0.74
|*dB/dt*| mT/s	16.30	26.07	32.58	1.15	1.84	2.30	2.31	3.71	4.63

## Data Availability

Not applicable.

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
