# Peer review of "Occupational Exposure Assessment of the Static Magnetic Field Generated by Nuclear Magnetic Resonance Spectroscopy: A Case Study"

_ijerph, 2022, doi:10.3390/ijerph19137674_

Round 1
Reviewer 1 Report
This manuscript assessed the occupational exposure to the static magnetic field generated by nuclear magnetic resonance spectroscopy, particularly for workers with active implanted medical devices (AIMD). Here, I found some comprehensive studies (Anna Sannino et al., 2017. Frontiers in Public Health, DOI: 10.3389/fpubh.2017.00344), (Kristel Schaap, et al., 2014. The Annals of Occupational Hygiene. 59, 817-820), that reported the exposure assessment of workers in magnetic resonance environment. The results obtained in this study seem to lack novelty. But, the results provided valuable occupational insights concerning NMR risk assessment. Some contents are suggested to be improved or better expressed.
Specific comments:
1. Line 118, “some measurements were repeated a few times to check the repeatability and the reproducibility of the measurements…”. But in the Results section, such as in Table 3-5, I can't learn whether the values presented in the tables were the mean or calculated only once, then I suggested you could provide additional tables in a supplemental file, where presented the values of repeated measurements, and their variable coefficient (CV%) or P values by ANOVA.
2. Line 134, I am not very sure if the Equation here should be evaluated by the weighted peak (WP) approach, as recommended by ICNIRP and by the 2013/35/EU Directive for non-sinusoidal signals (e.g., motion-induced electric fields).
3. Line 184-185, “… the body of a worker walking along a linear trajectory parallel to the x-axis starting close to the NMR spectrometer (x = 0 cm) to the console position (x = 270 cm)…”. I suggested adding MOD4, which is the current MOD2 starting to the console position to the NMR spectrometer, and it is also quite important in a real scenario.
4. In the Discussion section, the first three paragraphs are important but a bit stereotyped in the description of occupational exposure in NMR environments. More importantly, you need to have more words on your results and link them to previous literature, such as quite similar between Fig. 3(b) and 3(c), but they were designed in totally different models. Also, little discussion on tables 3-5.
5. This study would be more convincing if patients, volunteers, or live animals were involved.
Author Response
Responses to the Reviewer 1
Thank you very much for your comments. Following your remarks, we revised our manuscript and below you can find a point-by-point response to your criticisms and suggestions:
- Line 118, “some measurements were repeated a few times to check the repeatability and the reproducibility of the measurements…”. But in the Results section, such as in Table 3-5, I can't learn whether the values presented in the tables were the mean or calculated only once, then I suggested you could provide additional tables in a supplemental file, where presented the values of repeated measurements, and their variable coefficient (CV%) or P values by ANOVA.
Schaap's study (2014) had already been cited, now we added Sannino's one (2017); both studies are related to the movement of workers in MRI environment.
We implemented Materials and Methods Section by adding the number of recorded measurements. In addition, in results Section we added Figure 1 (b) that depicts the mean B values recorded at different heights from the ground plane and standard deviation.
“The gaussmeter was programmed to take instant B values every 10 seconds during 1-minute recording. The magnetic field |B| for each of the different measurements was estimated by the mean and the standard deviation (Mean ± sn-1) of the stored values.”
- Line 134, I am not very sure if the Equation here should be evaluated by the weighted peak (WP) approach, as recommended by ICNIRP and by the 2013/35/EU Directive for non-sinusoidal signals (e.g., motion-induced electric fields).
We added in the Discussion section the following sentence about this issue:
“A complete discussion about both controlled and uncontrolled exposure should include implications of higher spectral components: in this case, a weighted peak approach should be used [18]. However, as the PNS-based basic restrictions for all tissues in controlled exposure condition are constant over a large frequency range, there is no need for spectral weighting [18].”
- Line 184-185, “… the body of a worker walking along a linear trajectory parallel to the x-axis starting close to the NMR spectrometer (x = 0 cm) to the console position (x = 270 cm)…”. I suggested adding MOD4, which is the current MOD2 starting to the console position to the NMR spectrometer, and it is also quite important in a real scenario.
Thank you for your suggestion. We added the results relative to the path from the console position to the NMR spectrometer with walking speed model MOD2 (Figure 3 d):
“Figure 3 d) shows results relative to the walking speed model MOD2 for a walking path starting from the console position (x = 270 cm) to the NMR spectrometer (x = 0 cm).”
Data relative to the peak value of the calculated exposure parameters for the walking speed model MOD2 along the path starting from the console position to NMR spectrometer, are not reported since they are equal to the peak values obtained for the walking speed model MOD1 along the reverse path. This because in the proximity of NMR spectrometer, where the magnetic field is higher, the walking speed is equal to vmax in both cases.
We explained this in the text.
- In the Discussion section, the first three paragraphs are important but a bit stereotyped in the description of occupational exposure in NMR environments. More importantly, you need to have more words on your results and link them to previous literature, such as quite similar between Fig. 3(b) and 3(c), but they were designed in totally different models. Also, little discussion on tables 3-5.
In the Discussion Section we linked the obtained results to ones reported in literature:
“As expected, the data revealed that the higher B levels, corresponding to the highest exposure values, were recorded in the vicinity of the spectrometer. This result is similar to the one reported in [37].”
Moreover, we added some discussion about the similarity of Figure 3b and 3c:
“Observing the Figure 3 a-d, the maximum value of induced electric field is obtained in all the positions close to the NMR spectrometer, especially in the case where the worker moves away or approaches the NMR with high speed (Fig. 3a and Fig. 3d). In Figure 3 b and 3 c |E| has similar trends, despite referring to different walking speed models. This fact can be explained considering that both MOD2 and MOD3 an initial acceleration part followed by a constant speed part. However, in the first part of the worker path MOD3 includes higher speed with respect to the MOD2 so the values of |E| for this case are higher up to x= 180 cm. Then, MOD3 ends with a deceleration in the area with low magnetic field (from x=180 cm to x=270 cm) where, on the other hand, MOD2 has a high-speed equal to vmax; hence, in this final part, the values of |E| for the MOD2 are higher.”
We also added little discussion on tables 3-5:
“Another relevant finding is the strong dependence of the induced electric field results from the walking speed models: as it is possible to note from Tables 3-5, all the exposure parameters are very higher for the walking speed model MOD1 which considers a constant speed along the entire path. This fact implies that it is need to a deep knowledge of the worker movements to obtain an accurate estimate of the exposure parameters. A constant walking speed model can be used only for the worst-case scenario but it does not permit to obtain an exhaustive and realistic estimation of the exposure.”
- This study would be more convincing if patients, volunteers, or live animals were involved.
Thank you very much for your suggestion. In the next future, it is our intention to implement this study by using pocket dosimeters, which means they don’t hinder the operators’ movements. However, NMR spectrometers are used by researchers and no patients are involved during routinely operations.
Reviewer 2 Report
As the authors stated the aim of the study was the evaluation of the static magnetic field around an NMR spectrometer (300 MHz, 7 T) and, starting from these measurements, the theoretical evaluation, made by using developed software, of the time-varying magnetic fields due to the movement of workers. In this way, the evaluation of worker exposure, as risk assessment, is done by considering the induced electric field |E| and the time derivative of magnetic flux density |dB/dt| for providing safe working procedure. Particular attention is paid to workers with active implanted medical devices (AIMD) as a particularly sensitive risk group that must be protected against the dangers caused by the interference of electromagnetic field.
The literature about workers risks evaluation on Nuclear MR (NMR) spectrometer is poor.
This initial study is well structured and experimentation adequately designed in order to achieve the above defined aim. It contributes to the body of knowledge.
There is a discussion upon some limits of the study and additional research in order to enhance the NMR risk assessment.
However, the authors fail to elaborate more in detail future research work, and it is recommended to address it in the revised version.
Author Response
Responsens to the Reviewer 2
Thank you very much for your comments. Following your remarks, we revised our manuscript and below you can find a point-by-point response to your criticisms and suggestions:
However, the authors fail to elaborate more in detail future research work, and it is recommended to address it in the revised version.
In the Conclusion Section we elaborated more in detail the future research study:
“Epidemiological evidence on the potential effects of occupational exposure to static magnetic field is not considered probative of noxiousness. Studies in this research field are scarce. In the next future, it is our intention to implement this study by using wearable pocket dosimeters, which means they don’t hinder the operators’ movements. In this manner, it will be possible to gather a wide statistical pool of data to assist studies on health hazards and the need to develop behavioral rules for workers, especially when they move near the NMR spectrometer, and for workers with AIMD.”
Reviewer 3 Report
Interesting article reporting the outcome of a study mapping the magnetic field near a NMR spectrometer and investigating exposure levels of resulting fields on workers.
The results are interesting and support previous understanding.
I have the following comments/suggestions to improve the article:
- It would be useful to show a diagram illustrating the assumed room structure, centre of coordinate system, and assumed worker path in the introduction or method section.
- It would give the article more grounding if the assumptions used to produce equation (1) are stated explicitly, and indicate how this is related to Maxwell's curl equation or Faraday's law. In particular maybe indicate how C is related to physical parameters (such as conductivity, permittivity, shape/size of the object).
- It would be important to indicate how C is calculated and what are the assumed parameters for the workers (conductivity, permittivity) and their uniform in the calculations.
- Also in line 140, please confirm how T/s.m should be written (is it T/(s.m))?
- Also in general, use "with respect to .." in the text.
Author Response
Responsens to the Reviewer 3
Thank you very much for your comments. Following your remarks, we revised our manuscript and below you can find a point-by-point response to your criticisms and suggestions:
- It would be useful to show a diagram illustrating the assumed room structure, centre of coordinate system, and assumed worker path in the introduction or method section.
We added in Figure 1 a box to better explain the room structure, center of coordinate system and axis. Moreover, we added a box for each part of Figure 3 to indicate the walking speed model and the path referring to.
- It would give the article more grounding if the assumptions used to produce equation (1) are stated explicitly, and indicate how this is related to Maxwell's curl equation or Faraday's law. In particular maybe indicate how C is related to physical parameters (such as conductivity, permittivity, shape/size of the object).
We better explained the derivation of the Equation 1:
“|E| was estimated using the analytical model proposed in ICNIRP guidelines [18]. Starting from the Faraday’s law, which indicates that the induced electric field is directly related to the change of the magnetic flux through the body, and resolving it for a body cross-section perpendicular to the magnetic field we obtained the following equation: …”
- It would be important to indicate how C is calculated and what are the assumed parameters for the workers (conductivity, permittivity) and their uniform in the calculations.
We better explain what is C (now indicated as geometric factor) and how is calculated in this work:
“C is a geometric multiplier depending on the size and the shape of the body as well as the direction and distribution of the magnetic field. The geometric multiplier can be deter-mined considering an elliptical or circular cross-section perpendicular to the magnetic field [31]. In this work we considered an elliptical section with a= 0.4 m as the semi-major axial length and b= 0.2 as the semi-minor (C=0.16 Vm-1 per Ts-1).”
- Also in line 140, please confirm how T/s.m should be written (is it T/(s.m))?
We corrected it: C=0.16 Vm-1 per Ts-1
- Also in general, use "with respect to .." in the text.
We corrected them